# Mobility Restrictions and E-Commerce: Holistic Balance in Madrid Centre during COVID-19 Lockdown

**Rafael Villa [1],\* and Andrés Monzón [2]**

1 School of Technology and Science, Camilo José Cela University, 28692 Madrid, Spain
2 Transport Research Centre (TRANSyT), Universidad Politécnica de Madrid, 28040 Madrid, Spain; andres.monzon@upm.es
\* Correspondence: rvilla@ucjc.edu

**Abstract:** COVID-19 has brought about a substantial change in urban mobility, as well as an unprecedented increase in e-commerce throughout the world due to the emergence of new ways of shopping and consumption habits. In this context, urban logistics plays a crucial role in the triple bottom line of sustainability. The present document establishes a holistic vision of the problem aiming to (i) measure and compare the traffic generated in the Madrid Central area (low-emission zone) during the periods before and after the pandemic, and (ii) quantify e-commerce orders made by residents, as well as the Light Commercial Vehicles (LCV) required to deliver these parcels, measuring their environmental impact. The results show that road traffic in the Madrid Central area decreased by approximately 2/3 compared to normal levels and 1/2 in the case of LCVs. With regards to e-commerce, the number of parcels delivered doubled. This fact entailed an increase in the number of LVCs dedicated to package delivery in the central district and more pollution, but to a lesser extent than the growth of e-commerce. The challenge faced by urban logistics in the post-Covid era is managing to blend new mobility within large cities with the high volumes of e-commerce deliveries demanded by residents.

**Keywords:** city logistics; last-mile delivery; sustainable development; e-commerce; COVID-19; environmental economics; sustainable transport



## 1. Introduction

Habitability, sustainability and competitiveness; these are the main challenges faced by large cities throughout the world. Their prioritization translates into improved quality of life for inhabitants and facilitates development from the three perspectives of sustainability: economic, social and environmental. There is a clear consensus among the main stakeholders of smart cities that human welfare and needs should be the starting point for a city's development, always taking into account sustainability criteria.

According to the European Commission (European Commission 2021), the large majority of European citizens live in an urban environment, and over 60% live in urban areas with over 10,000 inhabitants. Urban mobility accounts for 40% of all road transport $CO_2$ emissions and up to 70% of other transport contaminants. In cities like Rome (166), Paris (165) and London (149), traffic congestion causes residents to lose a significant number of hours on the road (INRIX 2020). The challenge for local administrations lies in reducing this traffic congestion in order to improve the habitability and competitiveness of their cities (Demir et al. 2015).

Within the context of urban goods distribution, globalization and e-commerce have generated exponential growth in road transport by enabling the development of an open market where products can be purchased from any location. Goods travel throughout the world and most are delivered in cities. This effect has been compounded by the COVID-19 crisis, as consumers have had to adopt new ways of shopping and new consumption habits, leading to an increase in the percentage of users who purchase physical products online. The number of trucks and vans is increasing due to the rising popularity of e-commerce

and the desire for faster deliveries (Savelsbergh and Woensel 2016), which, in turn, has led to more frequent and split deliveries in residential areas.

This increase in the number of transport vehicles in cities translates directly into greater congestion and accidents (social), more air pollution and noise (environmental) and higher logistics costs, with a subsequent increase of product prices (economic). Each city has attempted to implement its own solutions, resulting in initiatives that have generally been suboptimal at addressing this triple balance (Macharis and Melo 2011). In late 2020, the European Commission presented a sustainable and smart mobility strategy (European Commission 2020) which defined a roadmap of 82 initiatives grouped into three main pillars: digitalization, resilience and greening of mobility, in terms of both individuals and goods. This includes an exhaustive set of measures for goods transport, including weight reduction in road transport, the definition of specific plans to achieve sustainable urban logistics, and greater use of intermodal transport, favoring the use of railways and waterways, both inside and outside cities.

In order to tackle this challenge, cities must face the difficult task of promoting systems of urban goods distribution that are environmentally friendly as well as sufficiently efficient to satisfy both society and logistics businesses. It is important to highlight that sustainable development objectives can be pursued through measures that are occasionally contradictory and generate a different impact based on the affected stakeholders (Gatta and Marcucci 2014). The new challenge for city logistics lies in finding solutions that are capable of absorbing an increase in urban transport of goods derived from new consumption patterns while, at the same time, minimizing the associated social and environmental impact.

Accepting the radical changes that the emergence of COVID-19 has brought to our society in most fields, the present document aims to compare the pandemic's repercussions on traffic, e-commerce and urban logistics in the central district of the City of Madrid (Madrid Central area). More specifically, we aim to answer the following questions:

- What has been the impact on city traffic of the mobility restrictions imposed due to COVID-19?
- How has the demand for e-commerce parcels evolved before and after the pandemic?
- What have been the implications of this increase in e-commerce for urban logistics and the environment?

## 2. Literature Review

*2.1. Urban Logistics: Context, E-Commerce and and Measures in New Scenario*

2.1.1. Urban Movement of Goods

The movement of urban goods is essential for economic vitality (Allen et al. 2000; Muñuzuri et al. 2005) and key for industrial, commercial and leisure activities which, in turn, are vital in wealth generation. Following the ideas presented by various authors (International Conference on City Logistics et al. 2004; González-Feliu et al. 2012; Cattaruzza et al. 2017), the movement of goods within cities can be grouped into three main categories: (i) movement between businesses -B2B-, movement to end consumers -B2C- and urban management movement. Figure 1 shows these types of movements of goods and their main organization modalities.

Of total urban traffic, the distribution of urban goods is responsible for approximately 15% on a typical city (Dablanc 2011). Moreover, it involves other activities requiring greater use of space in cities: loading-unloading, storage, etc. Within urban logistics, different studies in French cities have determined that 46% of the total urban movement of goods is related to B2C commerce (Cattaruzza et al. 2017).

This paper addresses the transport of goods related to B2C commerce (ECM), specifically, urban logistics derived from e-commerce.

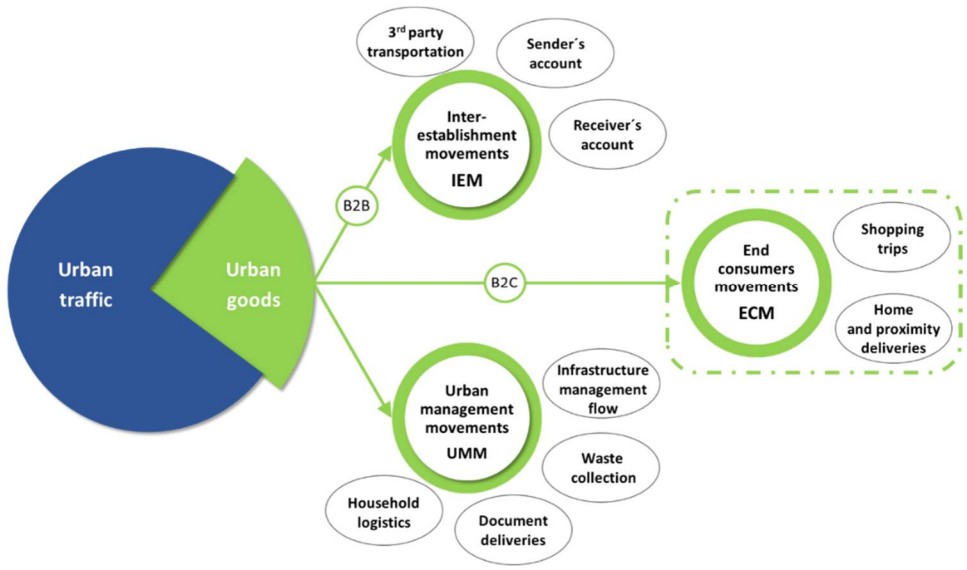

**Figure 1.** Classification of urban goods movement. Source: adapted from (Cattaruzza et al. 2017).

### 2.1.2. E-Commerce and Urban Logistics

Today, urban consumers can purchase everything they need online. In early 2020, the global population stood close to 7.75 billion inhabitants, of which 4.54 billion had used the Internet at least once, representing a penetration of 59%. Within the group of Internet users, 74% had made an online purchase during the studied period (We Are Social and Hootsuite 2021).

This continued growth of e-commerce throughout the world has accelerated in 2021, as COVID-19 has rewritten the rules of the retail sector. Between January 2019 and June 2020, retail platforms experienced an extraordinary increase in global traffic. The websites of retail businesses received nearly 22 billion visits in June 2020; a 35.5% increase year-on-year (Statista 2020a). In the United States, the share of e-commerce in total retail sales rose from 11.8 to 16.1% between the first and second quarters, and in the United Kingdom from 20.3 to 31.3%. In the EU-27, retail sales via mail order houses or the Internet in April 2020 increased by 30% compared to April 2019, while total retail sales diminished by 17.9% (OECD 2020a).

The reasons are evident: lockdown measures have driven new consumers to pursue online channel in order to avoid busy physical stores, and the shopping frequency of previous cyber-customers has increased and a multitude of businesses which did not yet have an online presence have launched such initiatives. E-commerce has become the only feasible option for many traditional brick-and-mortar stores during the pandemic, and has demonstrated its resilience by meeting growing consumer demand and ensuring the provision of essential goods and services, e.g., by posting products on social media sites and ordering product pick-up or delivery services (Koch et al. 2020; E-Commerce Europe 2021). However, the effect of the COVID-19 crisis on e-commerce has not been uniform across product categories or sellers. While the impact of COVID-19 on several categories has been considerable, it has had a much smaller impact on other products. Items related to food, fashion, electronics, beauty and household were the best-selling products, while others, such as tourism and airlines, have collapsed (OECD 2020b).

Amidst the unstoppable growth of e-commerce, while electronic transactions travel through data networks, the physical products being purchased still need to be transported and delivered to end consumers. During the first months of the pandemic, transportation and distribution of goods became one of the main causes of disruptions in the supply chain and affected the supply of essential items (Ivanov 2020; Linton and Vakil 2020). More people living in cities and simpler transactions for consumers translate into a higher frequency of deliveries and more vehicles on the road (Crainic et al. 2004;

Cardenas et al. 2017). This increase in e-commerce has resulted in increased pressure on last-mile logistics (Srinivas and Marathe 2021).

In addition, the on-demand economy and its instant deliveries have driven new consumer habits (Dablanc et al. 2017), where q-commerce (quick-commerce) has emerged as a new model within e-commerce, based on speed, convenience and customer care. Users value fast deliveries, being able to choose among different delivery options and being kept informed about the status of their orders. Regarding the evolution experienced by commerce in cities, Figure 2 shows the key characteristics of various generations of B2C commerce. Four new characteristics related to urban distribution are added to those previously established by Delivery Hero: Quick Commerce: Pioneering the Next Generation of Delivery (2020).

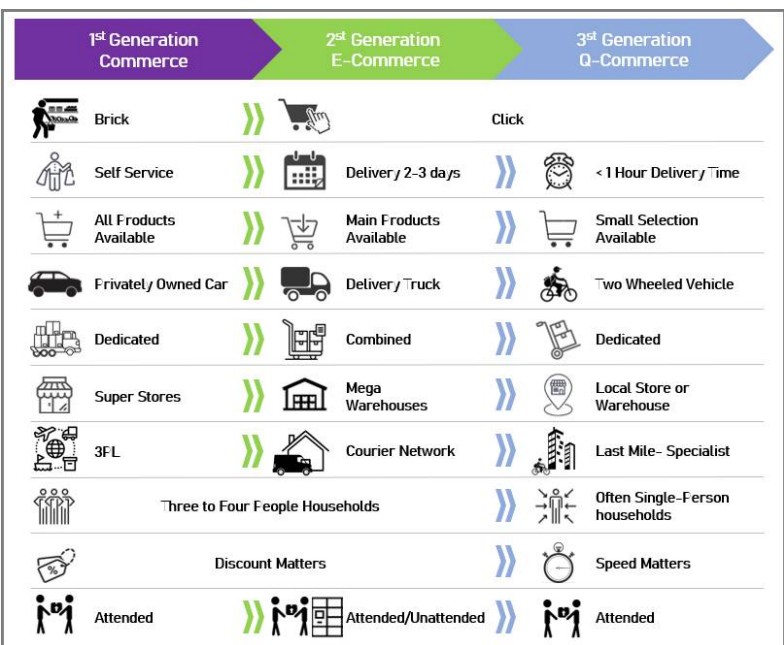

**Figure 2.** The evolution of quick commerce. Source: adapted from (Delivery Hero: Quick Commerce: Pioneering the Next Generation of Delivery 2020).

Figure 2 shows the evolution of consumer purchasing habits and the response by logistics operators to satisfy this demand. In the first generation of commerce, customers were the ones who visited physical stores, with a large product offering, to make their purchases. In q-commerce, purchases take place through a click, delivery time is a key variable and the product range is limited, since there are numerous online stores.

The logistics response to these shopping habits is very different. The first scenario deals with large volumes, employs large warehouses, optimizes loads and its essential element is cost. On the other hand, q-commerce volumes are small, operates through microhubs, response time is the key consideration and has couriers specializing in last-mile operations.

Q-commerce accentuates the difficulties already faced by e-commerce urban distribution: small volumes, more delivery addresses, higher resupply frequencies, lower stock levels, reduced optimization of vehicle loads and just-in-time deliveries (Lebeau and Macharis 2014). All of these elements entail an increasing dependence on urban roads and a need to find solutions for urban logistics. In this new context, lockers, collection points and mobile warehouses can have a positive impact from various perspectives and for all stakeholders involved in urban logistics by reducing the number of trips, failed deliveries and vehicles required. (Viu-Roig and Alvarez-Palau 2020).

### 2.1.3. Measures and Solutions to Improve Urban Logistics

In recent years, numerous initiatives have sought to minimize the negative effects of urban distribution and create the basis for a more solid and circular future economy where resources are employed more sustainably.

Different authors have classified and organized the significant number of measures proposed to improve city logistics into various categories (Russo and Comi 2011; Browne et al. 2012; Stathopoulos et al. 2012; CIVITAS 2015; Macharis and Kin 2017; Ranieri et al. 2018). However, the literature has mainly focused on the perspective of local authorities and political decision makers, despite the key role played by the private sector in many of these measures. Macharis and Kin (2017) focus on measures which explicitly include responsible stakeholders acting in city logistics. They classify these measures according to the so-called "four As": (i) awareness, (ii) avoidance, (iii) acting and (iv) anticipation of new technologies (see Table 1).

**Table 1.** Measures and solutions to improve urban logistics.

| | Type of Measure | Measure | Examples |
|---|---|---|---|
| **Public intervention measures** | Regulatory measures | Temporary access restrictions | Delivery restrictions during the day<br>Silent deliveries at night |
| | | Parking regulations | Loading and unloading restrictions<br>Vehicle parking reservation systems<br>Shared time in parking spaces |
| | | Environmental restrictions | Emissions standards and restrictions related to motors<br>Noise programs/regulations<br>Low emission areas |
| | | Access restrictions by size or load | Weight restrictions<br>Vehicle size<br>Load factor restrictions |
| | Market-based measures | Pricing (tolls, congestion tariffs and parking fees) | Road use tolls<br>Congestion tariffs<br>Parking fees |
| | | Taxes, tax breaks and incentives | High taxes for polluting vehicles<br>Subsidies for purchase of electric vehicles<br>Tax exemptions for electric vehicles |
| | | Negotiable permits and mobility credits | Purchase and sale of load transport services<br>Mobility credits in city centers |
| | Infrastructure and land use | Adaptation of street loading/unloading areas | Providing space on pavement for parking and loading activities |
| | | Building codes and construction regulations | New commercial premises providing adequate space for goods handling |
| | | Nearby delivery areas | Providing loading areas at public or private parking, empty areas, etc. |
| **Initiatives by urban logistics stakeholders** | Technological innovation measures | Innovation in vehicles | Electric vehicles<br>Unmanned vehicles: drones and terrestrial autonomous vehicles |
| | | Delivery points | Mailboxes for parcels<br>Smart lockers<br>Collection points |
| | | Advanced algorithms and optimization | Integrated inventory management<br>Task-courier matching<br>Route optimization<br>Data-driven demand forecast |
| | | Collaboration in urban logistics | Order or load capacity exchange<br>Collaborative local deliveries<br>Collaborative storage<br>Collaborative load sending |
| | Infrastructure and logistics systems | Urban infrastructure and logistics installations | Urban distribution centers<br>Microhubs<br>Consolidation of multiple operators |
| | | Urban logistics systems | Systems for underground transport of goods<br>Deliveries through public transport<br>Night distribution |

Source: own elaboration, based on (CIVITAS 2015; Macharis and Kin 2017; Ranieri et al. 2018).

These measures and initiatives are listed below, grouped into two main perspectives:

- Public intervention measures:

    (a)  Regulatory measures.
    (b)  Market-based measures.
    (c)  Infrastructure and land use planning

- Initiatives from players participating in urban logistics:

    (d)  Technological innovation measures
    (e)  Infrastructure and logistics systems

Therefore, there is no single solution for urban logistics; all measures work as levers that must be applied based on the characteristics and circumstances of each particular city and taking into account the interests of all stakeholders.

The remainder of this document is structured as follows: Section 3 describes the methodology used in the research. Sections 4 and 5 present the case study and results of the methodology for the Madrid Central area. Section 6 discuss the results and, lastly, Section 7 presents the conclusions and possible areas for further research.

### 3. Methodological Framework

To address the objectives described, the research is developed from two connected perspectives. First, from the descriptive side, traffic in the Madrid Central area is measured before and after COVID-19. Second, the effects of the pandemic on e-commerce are estimated, along with the impact on urban distribution of goods. The whole procedure is shown in Figure 3.

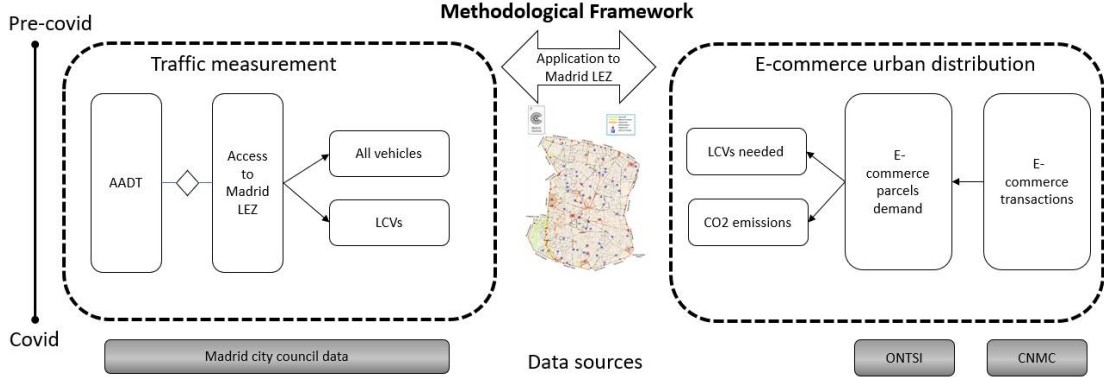

**Figure 3.** Methodological framework.

The first step was to measure the Annual Average Daily Traffic (AADT) and number of vehicles that accessed the Madrid Central area during the analyzed periods. The calculation of AADT was based on the information collected daily by City of Madrid through electromagnetic coils embedded into the pavement, which constantly quantify the passage of vehicles, thus enabling monitoring of traffic conditions in a given stretch of road. The measurement of vehicles accessing the Madrid Central area was based on data received from cameras placed on access points to monitor any registration plates entering it. The City of Madrid has installed 115 of these cameras to track vehicle access.

The second step consisted in analyzing the evolution of e-commerce in the studied periods through changes in consumption behaviors caused by the outbreak of COVID-19. The data was obtained from Spain's National Authority for Markets and Competition (CNMC). Quarterly e-commerce statistics take into consideration e-commerce (business volume and number of transactions) carried out using bank payment cards corresponding to the collaborating Spanish payment entity: Sistema de Tarjetas y Medios de Pago S.A. The products considered in this research are goods purchased through e-commerce and requiring physical distribution.

In the case of the Madrid Central area, the calculation of demand for e-commerce parcels considered home delivery as the most feasible and commonly used option. The calculation of this potential daily demand for e-commerce parcels was estimated through the chain-ratio method proposed by Kotler and Keller (2012), which multiplies a base number by several adjusting percentages to estimate the target demand.

The formulation would be as follows:

$$\text{Daily demand for ecommerce parcels by residents (D)} = A \times P1 \times P2 \times P3 \times P4 \times P5 \tag{1}$$

where:

*A = residents over 16 years of age*
*P1 = % average of residents over 16 years of age who use the Internet*
*P2 = % average of residents over 16 years of age who use the Internet and shop online*
*P4 = % average of residents over 16 years of age who use the Internet and shop online daily*
*P5 = % average of residents over 16 years of age who use the Internet and shop online daily for products that are physically delivered*

Resident data were obtained from City of Madrid public data repository, whereas data related to e-commerce came from the National Observatory for Telecommunications and the Information Society (ONTSI 2020), through its report "B2C e-commerce in Spain in 2019".

Lastly, once the number of online transactions had been defined, the next step was to calculate the vehicles required to deliver those orders, taking into account both the number of courier companies and the theoretical load of light commercial vehicles (LCV). The market share of each courier was obtained from the CNMC through its annual report on the evolution of the postal sector (CNMC 2019).

Likewise, the $CO_2$ emissions generated by last-mile e-commerce deliveries were estimated for the two periods analyzed. The calculation of emissions took into account previous estimations of $CO_2$ emissions per kilometer travelled. The reference value to calculate emissions is $kgCO_2$ per km, following data from the International Post Corporation (2018), DPDgroup (2019) and Deloitte (2020).

## 4. Case Study

Madrid is the largest city in Spain and the second largest in the European Union, with a population of 3,266,126. It is the core of Madrid Region, which has 6,663,394 inhabitants (Spanish Statistics Institute, INE 2020a).

Moreover, the Madrid Region has the highest GDP per capita in Spain and tenth highest in Europe, at over 35,000 euros per person in 2019 (INE 2020b). It is the seat of the main public institutions in the country and region, as well as the hub for political-administrative, financial and commercial activity. There are over 520,000 businesses in the region (16% of the country's total), but when narrowing the scope to those with over 500 employees, the percentage increases up to 40%.

Currently, the City of Madrid is divided into 21 administrative districts which, in turn, are comprised by 131 neighborhoods. Six of them form the Madrid Central area (see Figure 4), where we focused this research and which also make up the oldest part of the city. This area has a total surface of 523.73 ha and a population of 140,473 inhabitants as of 1 January 2020.

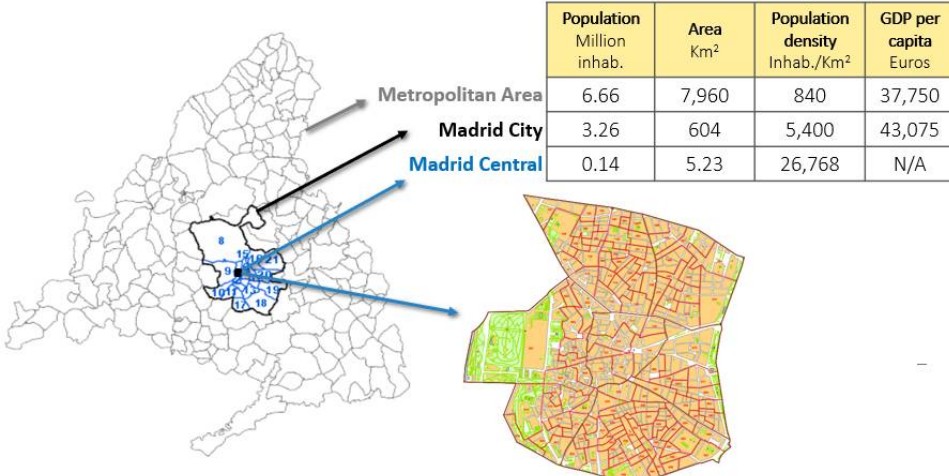

| | Population Million inhab. | Area Km² | Population density Inhab./Km² | GDP per capita Euros |
|---|---|---|---|---|
| Metropolitan Area | 6.66 | 7,960 | 840 | 37,750 |
| Madrid City | 3.26 | 604 | 5,400 | 43,075 |
| Madrid Central | 0.14 | 5.23 | 26,768 | N/A |

**Figure 4.** Region of Madrid, City of Madrid, and Madrid Central (adapted from City of Madrid website).

### 4.1. Urban Transport

Urban transport in the City of Madrid, of both passengers and goods, has been identified as one of the most complicated among large European cities due to the orography and historical evolution of its urban structure. In terms of urban morphology, the City of Madrid presents an irregular and radio-centric map, with narrow streets and closed construction plans combined with large squares and regular avenues created by the successive remodeling undergone by the city since the 16th century.

Regarding traffic and urban distribution, Madrid's vehicle pool has a high percentage of diesel vehicles as well as older models, with an average age of 9.3 years (Área de Gobierno de Medioambiente y Movilidad 2019). According to the Inventory of Atmospheric Pollutant Gas Emissions (Madrid City Council Environment and Mobility Office 2019), road transport accounted for 34.1% of total greenhouse gas (GHG) emissions. In late 2018, commercial and industrial vehicles older than 10 years represented 73.2% of the total (ANFAC 2019).

In Madrid, congestion related to urban logistics reached 38% (18 points more than in the rest of Spanish cities) and has been forecasted to rise up to 47% by 2025 (Madrid College of Economists 2020).

### 4.2. Madrid Central Area (Madrid LEZ)

A study analyzing 858 European cities concluded that the metropolitan section of Madrid was the urban area with the highest mortality related to nitrogen dioxide ($NO_2$) pollution in the continent. The study by ISGlobal (Khomenko et al. 2021) calculated that if all the analyzed cities reduced their concentrations of fine particles and $NO_2$ pollution to the levels recommended by the World Health Organization (WHO), they would prevent 51,000 premature deaths attributed to the former and 900 attributed to the latter.

In November of 2018, the City of Madrid defined a low-emission zone (LEZ) or "Madrid Central". This measure, known as "Madrid 360" since 2020, restricts the access of private vehicles to the central district of the capital in an effort to promote pedestrian mobility, bicycles and public transport. The only vehicles allowed are those belonging to residents, individuals with reduced mobility and security and emergency services. Logistics and distribution vehicles are allowed access to the 472 ha, but have been given a deadline to modernize their fleets.

The Madrid Central area is the core of the LEZ. Access regulations only allow eco-friendly vehicles with "0 Emissions" and "ECO" stickers, i.e., hybrid and electric vehicles, to drive and park in the area. Other vehicles can only access it if they are residents or to park in public parking or private garages.

## 5. Results

### 5.1. Calculation of Vehicle Traffic and Access to the Central District

Figures 5 and 6 show (i) traffic intensity and (ii) vehicle access to Madrid Central area for January–June in 2019 and 2020 (before and after COVID-19). The intensity, that is, the number of vehicles per hour, was registered at 177 measurement points distributed among the six neighborhoods of the Madrid Central area. Over 4,800,000 different measurements were recorded in all timeframes.

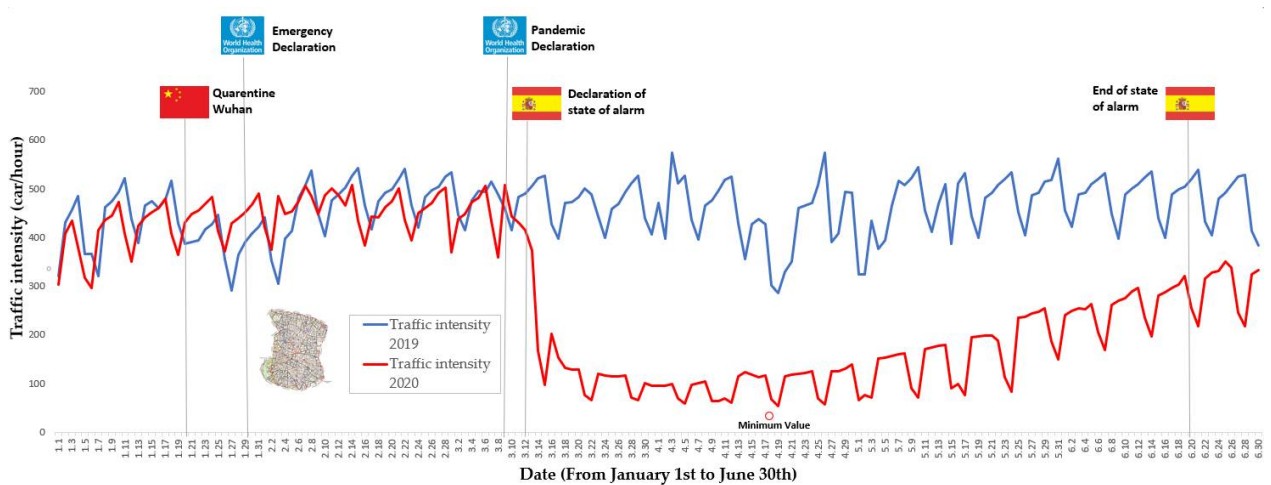

**Figure 5.** Traffic intensity 2019–2020 (Q1 and Q2: first and second quarter of the year).

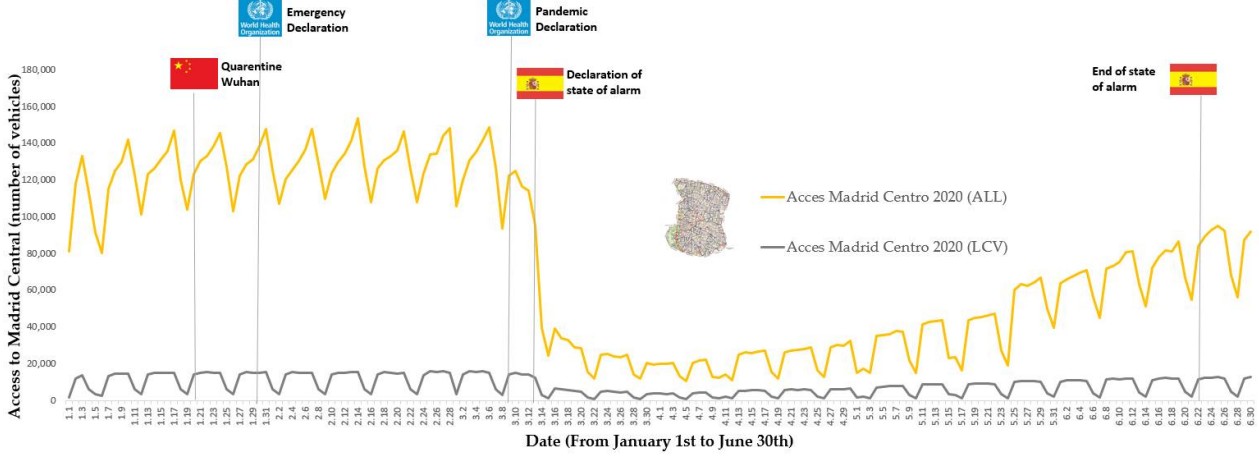

**Figure 6.** Vehicle access to the Madrid Central area in 2020 (Q1 and Q2).

As Figure 5 shows, the number of vehicles per hour was very similar during the first two months of both periods. The inflection point was 11 March, 2020, when the WHO declared a pandemic. Traffic intensity started to decrease considerably in the following days until, on 14 March, the Spanish Government declared a State of Alarm, limiting freedom of movement for citizens except for specific purposes and ordering most businesses to close, along with all leisure, education and cultural sites. On 15 March 2020, traffic intensity in the Madrid Central area was just 18.58% compared to the same day of the previous year.

Subsequently, in May 2020, traffic volume began to increase gradually until the end of June 2020. This increase corresponded to the progressive lifting of mobility restrictions in the City of Madrid. On 21 June, the State of Alarm was lifted, putting an end to the de-escalation process and bringing the country into the "new normal". Nevertheless, on the last week of June, the "new normal" of 2020 saw 35.8% less traffic in the city center than on the same week of 2019.



Figure 7 shows vehicle access to the Madrid Central area during the analyzed periods (distinguishing access for all vehicles and for LCVs) and how the evolution was virtually the same as for traffic intensity. The correlation between traffic intensity and vehicle access (all vehicles) was 0.932 for 2019 and 0.995 for 2020. Regarding LCVs access, during 2019, this represented 9.72% out of the total vehicles accessing the Madrid Central area. This percentage remained similar during the pre-Covid period of 2020 (9.33%) and later increased up to 14.8% from 14 March to 30 June. In other words, the reduction in mobility in the Madrid Central area was more notable for other vehicles than for LCVs.

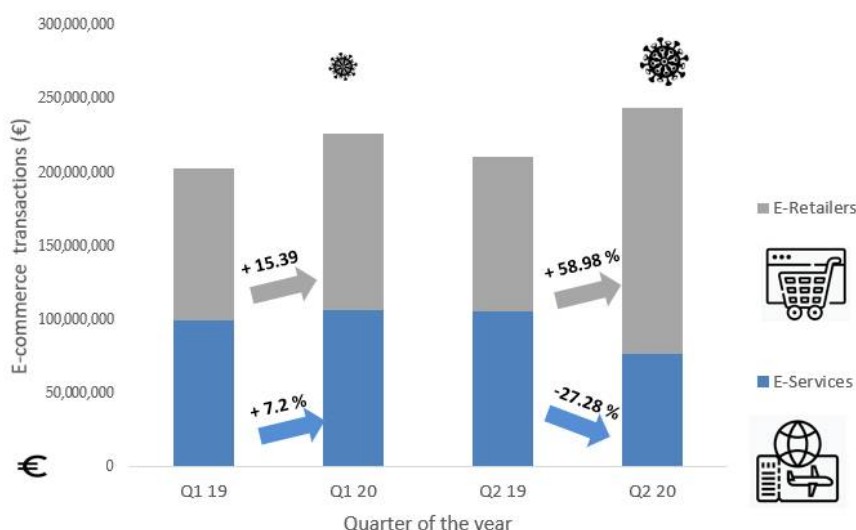

**Figure 7.** E-commerce transactions 2019–2020 (Q1 and Q2).

*5.2. Calculation of E-Commerce Transactions for Physical Goods in Spain Pre- and Post-COVID*

Concerning online sales, Figure 7 shows the volume of e-commerce transactions carried out by Spanish citizens throughout the analyzed period.

Focusing exclusively on e-retailers, or products requiring physical delivery, the increase in transactions reached 15.38% during the first quarter of 2020 (the last 15 days of this period correspond to the State of Alarm). In contrast, when comparing the second quarter of both years, the increase in physical goods purchased via e-commerce reached 58.97% during the first wave of the pandemic. Purchases at hypermarkets and supermarkets doubled and purchases of beverages, household appliances and audio-visual equipment tripled. Table 2 shows the 10 most popular product categories during the pandemic and their evolution throughout the 2014–2020 period. But not all sectors have experienced the same impact. Food, fashion, electronics, household products, beauty and parapharmacy have had remarkable growth while, for obvious reasons, tourism and airlines have been practically paralyzed.

**Table 2.** Top 10 e-commerce product categories with highest growth rates.

| ACTIVITY | 2014-2020 (Q2) | YOY Growth Rate (Q2: 2019-2020) |
|---|---|---|
| FURNITURE, LIGHTING AND HOME | | 318% |
| HOME APPLIANCES, VISUAL AND AUDIO PRODUCTS | | 310% |
| BEVERAGES | | 291% |
| TOYS AND SPORTS ITEMS | | 247% |
| HARDWARE, PAINTS AND GLASS | | 244% |
| OTHER NON-SPECIALIZED TRADE | | 243% |
| PERFUMERY, COSMETICS | | 224% |
| MEDICAL AND ORTHOPEDIC ITEMS | | 218% |
| HYPERMARKETS, SUPERMARKETS AND FOOD SHOPS | | 213% |
| BODY MAINTENANCE | | 203% |

Source: own elaboration.

*5.3. Calculation of Demand for E-Commerce Parcels*

Figure 8 shows an estimation of the daily demand for e-commerce parcels for residents of the Madrid Central area. The estimate distinguishes total demand by quarter for the period from January 2019 to June 2020. As shown, while the order volume increased slightly during 2019, the number of parcels delivered in the central district exploded after the start of the pandemic, almost doubling in number.

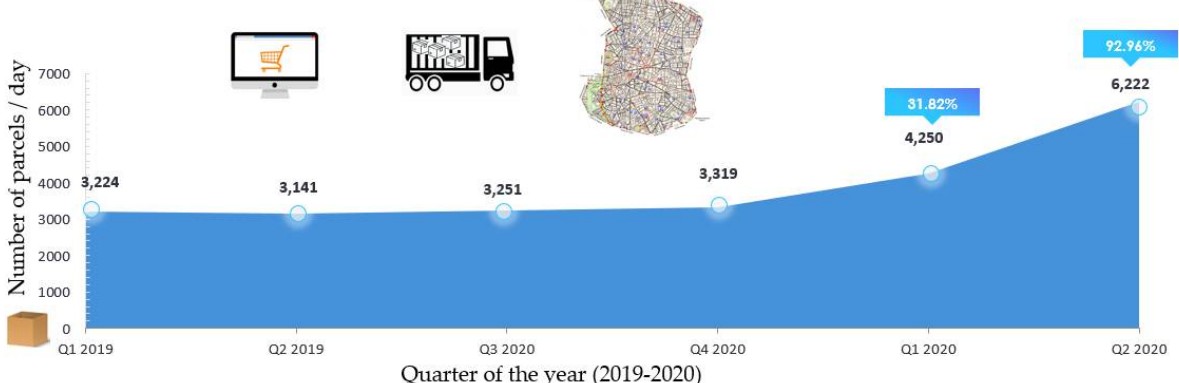

**Figure 8.** Daily online e-retail products 2019–2020.

As shown in Figure 9, the calculation of e-commerce demand begins with the total residents of the central district over 16 years of age and, applying the chain-ratio methodology, estimates the number of daily online shoppers for e-retailer products. The figure refers to Q1 2019.

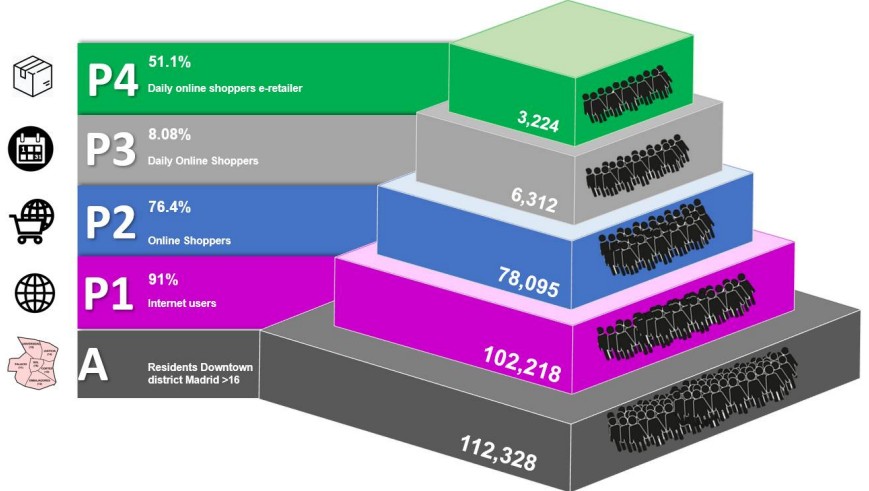

**Figure 9.** E-commerce factors explaining daily demand for residents.

*5.4. Calculation of Delivery Van Fleet*

The calculation of vans required to deliver e-commerce parcels takes into account current operations in last mile logistics in the City of Madrid. These follow the traditional scheme, with large sorting and delivery centers located in the outskirts, in towns like Coslada, San Fernando de Henares, etc. These are large-scale fulfilment centers handling significant volumes. From these warehouses, LCVs service the different urban centers through routes of approximately 80–120 km per day and vehicle, delivering 80–125 parcels each day throughout long delivery periods (Deloitte 2020). Higher traffic intensity entails lower values in this range whereas, with lower intensity (greater fluidity), couriers are able to deliver a larger number of parcels in each route.

Figure 10 represents express and parcel delivery market share, where 10 companies account for nearly 75% of the Spanish courier sector. The remaining 25% is divided among a large number of companies with a very small market share. Therefore, only companies dealing with significant volumes are able to optimize loads and routes simultaneously.

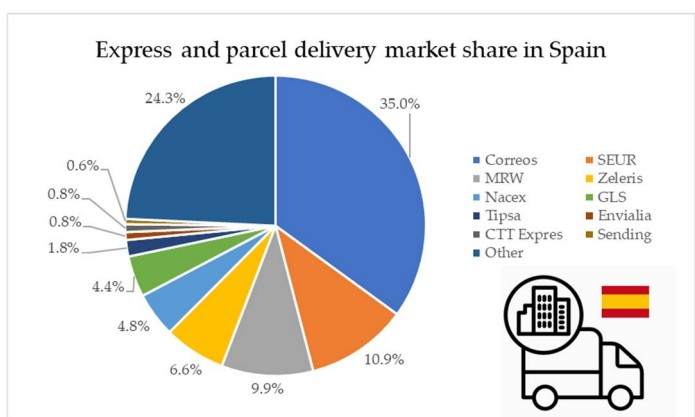

**Figure 10.** Market share: express and parcels delivery in Spain (source: CNMC 2020).

A larger number of couriers in the ecommerce market delivery brings down transport load optimization, since many operators will not have enough parcels to fully load their vans. In this scenario, they must either complete the route with less than their maximum capacity or combine it with other delivery areas.

Table 3 shows the minimum number of LCVs required to deliver the parcels requested by residents of the Madrid Central area, taking into account the market share of the courier sector and delivery productivity based on traffic (a value of 80–125 parcels/LCV is

considered following the traffic intensity explained in Appendix A). The figures represent daily data corresponding to the analysed period.

**Table 3.** Number of LCVs required to deliver parcels.

|  | Q1 2019 | Q2 2019 | Q1 2020 | Q2 2020 |
|---|---|---|---|---|
| Correos (35.0%) | 15 | 14 | 17 | 18 |
| SEUR (10.9%) | 5 | 5 | 6 | 6 |
| MRW (9.9%) | 4 | 4 | 5 | 5 |
| Zeleris (6.6%) | 3 | 3 | 4 | 4 |
| Nacex (4.8%) | 2 | 2 | 3 | 3 |
| GLS (4.4%) | 2 | 2 | 3 | 3 |
| Tipsa (1.8%) | 1 | 1 | 1 | 1 |
| Envialia (0.8%) | 1 | 1 | 1 | 1 |
| CTT Express (0.8%) | 1 | 1 | 1 | 1 |
| Sending (0.6%) | 1 | 1 | 1 | 1 |
| Other (24.3%) | >40 | >40 | >40 | >40 |

One relevant issue is the "Other" category, which groups nearly 25% of deliveries. This long tail (not quantified, since the breakdown was not available), represents a high number of LCVs carrying few parcels. In addition, this proportion of LCVs remained constant throughout the periods analyzed, since increasing the number of parcels also increased the number of delivery hours in the central district and load optimization of each LCV, but not the number of vehicles required.

*5.5. Calculation of the Fleet of Delivery Vans*

Considering LCVs emissions based on parcels delivered and kilometers driven in each daily route, the environmental impact derived from the delivery of e-commerce parcels during the periods analyzed, measured in kg of $CO_2$ equivalent, is shown in Figure 11. $CO_2$ emissions for Q1 of both years are very similar in line with the number of packages delivered. On the other hand, for Q2, the increase in e-commerce orders (+98%) translates into a higher number of emissions but, due to the higher delivery productivity during the COVID-19 period, the increase in $CO_2$ is lower (+43.1%).

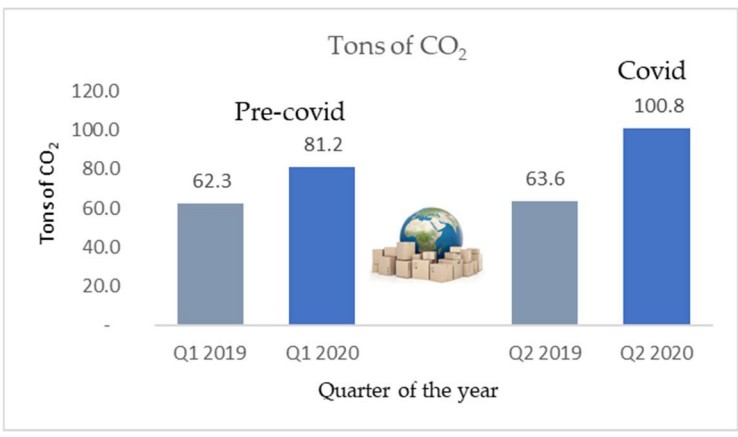

**Figure 11.** Tons of $CO_2$ emissions.

The reference value to calculate emissions is $kgCO_2$ per km, following the data from the DPDgroup (2019) and Deloitte (2020).

## 6. Discussion

The analysis of results seeks to answer in greater detail the questions posed at the start of the study. Table 4 reflects the number of vehicles per hour and access volume to the Madrid Central area for three specific time periods: January–June 2019 (pre-Covid), from January 1 to March 13 (pre-Covid) and from 13 March to 30 June (during Covid).

**Table 4.** Central district traffic statistics.

| Date | AADT (Vehicles/h) | | | Vehicle Access (Vehicles/Day) | | | Vehicle Access (LCVs/Day) | | |
|---|---|---|---|---|---|---|---|---|---|
| | **Mean** | **SD** | **% Mean** | **Mean** | **SD** | **% Mean** | **Mean** | **SD** | **% Mean** |
| (Q1+Q2) 2019 | 444 | 62 | - | 131,351 | 21,371 | - | 12,764 | 5580 | - |
| 01/01–03/13/20 | 439 | 48 | −1.05 | 125,375 | 15,596 | −4.55 | 11,415 | 4827 | −10.57 |
| 03/14–06/30/20 | 166 | 83 | −62.24 | 41,095 | 24,227 | −67.22 | 6079 | 3789 | −46.74 |

In Madrid, COVID-19 paralyzed all activities considered as nonessential and, therefore, movement was limited to these basic activities. On average, traffic intensity decreased by 62.24%. In turn, since the Government implemented the State of Alarm which locked down most of the population until the so-called "new normal" (11/05/2020), road traffic dropped, on average, down to 76.28%. Similar results were reported in the UK, where road traffic volumes fell by as much as 73% (Budd and Ison 2020), and in other cities around the world: New York (−74%), Barcelona (−73%), Milan (−74%), Stockholm (−48%) and Sao Paulo (−55%) (year-on-year traffic reduction between 16 March to 22 March 2020; Statista 2020b). In this context, all modes of transportation were affected and it would be interesting to know how citizens changed their daily commute preferences due to the healthcare crisis. In the case of car access volume, the reduction was similar for total vehicles (67.22%), but significantly lower for LCVs (46.74%). This observation is explained by the fact that, due to the state of alarm, access to the central district was restricted to key activities, including supply of essential goods and services, home delivery of food, healthcare services and the necessary industry to conduct key activities. Under normal conditions, urban distribution accounts for 20% of total traffic in Madrid (DGT 2020), but this percentage rose during the analyzed period, due to the decrease in general traffic when compared to LCV traffic reduction. Focusing only on the period of the state of alarm, overall vehicle access to the Madrid Central area fell by 82.18% and in the specific case of LCVs, by 67.38%.

In contrast, e-commerce transactions saw a significant increase during that same period, as consumers embraced new ways of shopping and adopted new consumption habits due to the lockdown. Comparing the second quarter of 2020 (amidst the pandemic) with the same period from the previous year highlights a doubling of e-commerce retail purchases for residents of the Madrid Central area (see Figure 8). It is important to note that this growth of e-commerce was not only due to an increase in shopping frequency by customers who already used the online channel, but also the emergence of new buyers who had previously been reluctant to make purchases through the Internet (ONTSI 2020). The necessity created by the limitations imposed forced these new customers to face that unknown barrier. All the signs seem to indicate that, once this obstacle has been overcome, most new customers will continue making purchases through the new channel.

This increase in e-commerce orders translated into a larger number of LCVs circulating through the city but, as a result of the reduced traffic, the number of parcels delivered in each route increased and less LCVs were required to absorb the increase (higher load optimization per LCV). Given the large number of couriers with a small market share, one alternative to consider would be the consolidation of these operators' e-commerce parcels through a microhub located in the central district. This could be implemented via public (microhub) and private (logistics operators) collaboration.

In order to better compare the main magnitudes of the study, Figure 12 shows the variations in the periods analyzed.

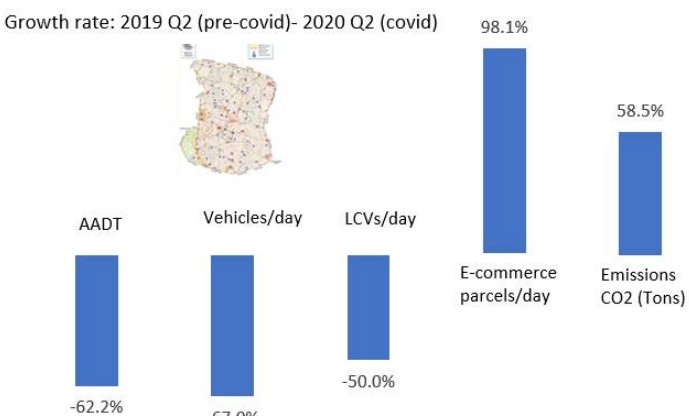

**Figure 12.** Variation in the analyzed magnitudes.

Lastly, from a sustainability standpoint, it is important to highlight the increase in $CO_2$ caused by the growth of e-commerce for residents of the Madrid Central area. Comparing the second quarter of both years, emissions rose by 50%, a smaller increase than that experienced by the number of parcels. A greater decrease in emissions should be sought through the use of other, cleaner means for last-mile deliveries (bicycles, delivery on foot, electric vehicles, etc.) or else through greater productivity in the LCV kilometer/delivered parcel ratio. The option of incorporating a microhub to consolidate parcels would also lead to a reduction in $CO_2$ emissions.

With regard to congestion, the increase in e-commerce parcels has meant an increase in vans in the central district, but in a significantly lower proportion than the increase of online orders. This is explained by three main reasons: (i) the reduction in traffic intensity increases the number of parcels delivered through each route, (ii) transport operators with a lower market share are able to load more parcels into each vehicle and (iii) the greater the demand for parcels in a given delivery area, the greater the possibility of delivering multiple orders in each stop.

## 7. Conclusions and Future Perspectives

This study describes the impact of COVID-19 on traffic in the urban center of a large city. In addition, it quantifies the demand for e-commerce parcels by residents of an urban center and estimates the impact on LCV traffic, considering its environmental repercussions. The results, applied to the central district of a city like Madrid, seek to answer the questions posed in the introduction.

First and foremost, road traffic in the Madrid Central area was directly affected by the lockdown measures. During the period of the pandemic analyzed (Q2 2019), economic activity entered a state of hibernation and mobility was reduced to essential activities, which reduced traffic to approximately 35% of normal rates.

The exception was e-commerce, where transactions for physical goods increased by 98% during this period, in line with online retail shopping behavior in other EU countries and the US (Eurostat 2021; BCG Global 2020). The growth of e-commerce caused an increase in the number of vehicles dedicated to transporting e-commerce orders, albeit in a notably smaller proportion than the increase in demand. Courier companies have found themselves in an ideal scenario with increased demand and empty streets, enabling them to make deliveries with very few limitations.

In this exceptional context, it would not be reasonable to apply public intervention measures, as these generally focus on decreasing traffic congestion and vehicle emissions under circumstances of traffic saturation, a situation which did not take place. $CO_2$ emissions related to e-commerce last-mile increased 43.1% during the pandemic period,

but this increase in $CO_2$ is no relevant if we consider the global reduction of all pollutant emissions in cities due the reduction in traffic and other activities. Average $NO_2$ levels during the week of 16–22 March went down by 41% in Madrid, 51% in Lisbon, 55% in Barcelona, 21% in Milan and 26–35% in Rome (Cheval et al. 2020). Therefore, environmental measures could focus on using innovative technologies: IoT (Internet of Things), big data, parcel lockers, electric vehicles, route optimization algorithms, collaboration among couriers and the use of urban distribution centers (Taniguchi et al. 2020).

Will the world after COVID-19 bring a new normality or a new reality? It is undeniable that, once the pandemic is over, the world will be substantially different in multiple aspects. Two such examples are those studied in this research: urban mobility and an increase in e-commerce. In this new, uncertain scenario, it will be essential to adopt measures that stakeholders can agree upon, in order to improve urban distribution in large cities from a sustainable perspective. Focusing on the environmental perspective, the increase in courier activities, added to the new consumption and mobility trends, highlight the need to promote improvements in the current models for urban transport and distribution of goods, including: public-private collaboration for retailers and for transport and logistics operators, environmentally friendly vehicles for city dwellers and raising e-commerce customer awareness and regulation (Russo and Comi 2020).

Inevitably, the study has several limitations, which provide valuable paths for future research. First, it would be relevant to have a complete picture of citizen mobility during the pandemic, that is, to know the exact percentage who used public transport, how many used their private vehicles and how many chose to move around on foot or by bicycle. This would provide an understanding of the transfers that took place between different modes of transportation. Obtaining this information for the post-Covid period could be very useful in defining better urban mobility and logistics policies in the future. Another essential element would be calculating the exact percentage of vehicles employed in urban logistics versus total traffic. Understanding this information and knowing delivery schedules would contribute to more efficient and sustainable proposals for urban logistics and traffic in big cities. Furthermore, it would be valuable to study how other modes of delivery (smart lockers, collection points, etc.) may contribute to improve economic aspects for couriers and social and environmental aspects for the city.

Lastly, this study could be extended from the perspective of city logistics operators, examining specific initiatives to improve urban distribution of goods in the context analyzed. Likewise, it would be useful to extend this study to evaluate the economic, social and environmental impact of the pandemic on both road traffic in general and urban distribution of goods in particular. Moreover, this analysis requires more detailed studies considering the new post-Covid reality, where mobility in the city will be different and new consumer habits will require more resilient and efficient urban logistics. These aspects will be developed in future research.

**Author Contributions:** All authors contributed equally to the data preparation and the analysis and to the interpretation of results. All authors made major contributions to writing the manuscript. All authors have read and agreed to the published version of the manuscript.

**Funding:** This research received no external funding.

**Institutional Review Board Statement:** Not applicable.

**Informed Consent Statement:** Not applicable.

**Data Availability Statement:** Data related to E-commerce transactions can be found in CNMC database at http://data.cnmc.es/datagraph/. Data related to traffic measurement can be found in Madrid city council data at https://datos.madrid.es/portal/site/egob. Data for the calculation of the variables in E-commerce parcels demand can be found in ONTSI at https://www.ontsi.red.es/es/ontsi-data.

**Conflicts of Interest:** The authors declare no conflict of interest.

## Appendix A

- Productivity of LVCs in Madrid: 80–125 parcels/day
- Kms driven by LCVs in Madrid: 70–125 kms/day
- Emissions per van: 180–250 g $CO_2$/km

The values considered in the case of the Madrid Central area are as follows:

| Operational Aspect | Traffic: Dense Loading/Unloading: Difficult (Pre-Covid Period) | Traffic: Fluid Loading/Unloading: Easy (Covid Period) |
|---|---|---|
| Productivity LVCs/day | 80 | 125 |
| Kms driven LVCs/day | 72 | 90 |
| Emissions per LCV | 250 g $CO_2$/km | |

Interval of 80–125 parcels/LCV considers traffic intensity (Deloitte 2020; International Post Corporation 2018; DPDgroup 2019).

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
