# Peer review of "Mobility Restrictions and E-Commerce: Holistic Balance in Madrid Centre during COVID-19 Lockdown"

_economies, doi:10.3390/economies9020057_

Round 1

Reviewer 1 Report

Dear Authors,

thank you a lot for the interesting paper. You chose a very important topic. On all countries all over the world the pandemic caused many changes, for example in transport and shopping habits. In my country (Poland) in some cities, when everything was closed, almost no people, no cars on the streets, we could see wide animals in city centers. In many cities, where normally we have problem with smog, suddenly we had better air to breath. These are positive causes of this negative problem we have now.

I also wrote a paper about it and made a survey how shopping behavior and shopping structure had changed cause of pandemic. Personally I can say that when everything started I also started to buy many things on Internet that before I had preferred to go to stationary shopping. The same my friends. So the subject you chose is really important and up-to-date.

Reading you text I was thinking that your paper should have been submitted to Sustainability (MDPI journal). Because you include ecological and economical aspects of the transport changed caused by covid-19 and online shopping. But it is only my opinion.

Your paper is well constructed, includes all necessary section. In abstract all content is well summarized. The introduction presents the reason why authors wanted to present their results. Methodology and results are presented in reasonable and interesting way. All paper seems to be good written and reasonable. References relatively new and adequate to the presented text. However, I found some mistake that should be corrected. I think that paper is good and interesting for potential readers. It needs only some corrections.

So now elements that should be corrected:

  1. Key words: You applied to SI. I think you should include any keywords from those proposed in SI.
  2. Introduction: lack of aim of the paper, you contribution to science etc... You included aim etc… later in point 2
  3. Line 34: lack of dot at the end of the line.
  4. Line 74: “the following figure” better to change into “Figure 1”. The same later in the text. In this way the reader knows exactly where to search. Line 122-123 “Figure 2” instead of “following image”. Line 164: “Table 1”. Line 311: “Figure 5” instead of “the two following graphs”. Line 319: Figure 5 instead of the graph.
  5. Figure 1: Is it possible to enlarge letters? I have difficulty to read. And grey background does not help.
  6. Figure 2: Please, make a comment about this Figure.
  7. Line 177-185 should be in the introduction for me. he reader does not want to search for it.
  8. Figure 4: enlarge letters in the table.
  9. Figure 5: Axes titles, also other figures. I am the mathematician, I am very grumpy about such things.
  10. Explain somewhere Q1 and Q2. I guess what it means but not for everyone it will be clear.
  11. Line 329: “new normal.” – dot after quote.
  12. Line 348-358: poor format of the text (text alignment).
  13. Page 12: footnote. Check in the template if the footnote can be used.
  14. Table 4: this small drawing on left: is it necessary?

I chose "Reconsider after major revision" to see your progress. I hope you will correct it as fast as possible, so it can be considered by MDPI for publication.

Author Response

Response to Reviewer 1 Comments

Thank you for your comments. I really appreciate them and I think they make the paper improve remarkably

Point 1: Key words: You applied to SI. I think you should include any keywords from those proposed in SI.

Response 1: Totally agree. Environmental economics and sustainable transport keywords have been added. (Line 19-20)

Point 2: Introduction: lack of aim of the paper, you contribution to science etc... You included aim etc… later in point 2

Response 2: I have included this part at the end of the Introduction (Line 66-74).

The contribution to science: “, the present document aims to compare the pandemic’s repercussions on traffic, e-commerce and urban logistics in the central district of the city of Madrid” and  the specific objectives:

  • What has been the impact on city traffic of the mobility restrictions imposed due to Covid-19?
  • How has the demand for e-commerce parcels evolved before and after the pandemic?
  • What have been the implications of this increase in e-commerce for urban logistics and the environment?

Point 3: Line 34: lack of dot at the end of the line.

Response 3: Done. “The challenge for local Administrations lies in reducing this traffic congestion in order to improve the habitability and competitiveness of their cities (Demir et al., 2015).” (Line 35).

Point 4: Line 74: “the following figure” better to change into “Figure 1”. The same later in the text. In this way the reader knows exactly where to search. Line 122-123 “Figure 2” instead of “following image”. Line 164: “Table 1”. Line 311: “Figure 5” instead of “the two following graphs”. Line 319: Figure 5 instead of the graph.

Response 4:

”the following figure” changed to “Figure 1” (Line 85)

“following image” changed to “Figure 2” (Line 138)

“Table 1” is added. “They classify these measures according to the so-called “four As”: (i) awareness, (ii) avoidance, (iii) acting and (iv) anticipation of new technologies (see table 1). (Line 194).

“Figure 5” instead of “the two following graphs”. Done. “Figure 5 and figure 6 show……” (Line 358).

Line 319: Figure 5 instead of the graph. Changed to “Figure 5 show….” (Line 366).

Point 5: Figure 1: Is it possible to enlarge letters? I have difficulty to read. And grey background does not help

Response 5: Figure 1 has been modified with larger letters and the grey background has been removed. (Line 95)

Point 6. Figure 2: Please, make a comment about this Figure

Response 6: I have included the following comment:

Figure 2 shows the evolution of consumer purchasing habits and the response by logistics operators to satisfy this demand. In the first generation of commerce, custom-ers were the ones who visited physical stores, with a large product offering, to make their purchases. In q-commerce, purchases take place through a click, delivery time is a key variable and the product range is limited, since there are numerous online stores.

The logistics response to these shopping habits is very different. The first scenario deals with large volumes, uses large warehouses, optimises loads and its essential ele-ment is cost. On the other hand, q-commerce volumes are small, operates through mi-crohubs, response time is the key consideration and has couriers specialising in last-mile operations. According to a study by Deloitte for the Danish market, after the pandemic, convenience will overtake price as the top driver of online shopping, as consumers will become used to the channel and these habits will be ingrained (Deloitte, 2020).   (Line 163-174)

Point 7. Line 177-185 should be in the introduction for me. he reader does not want to search for it.

Response 7: Totally Agree. I have included this part at the end of the Introduction (Line 66-74).

Point 8. Figure 4: enlarge letters in the table.

Response 8: Figure 1 has been modified and larger letters included (Line 302)

Point 9. Figure 5: Axes titles, also other figures. I am the mathematician, I am very grumpy about such things.

Response 9:  Not grumpy, very useful. Figure5, figure 6, figure 7, figure 8 and figure 10 has been modified. Axes titles included

Point 10. Explain somewhere Q1 and Q2. I guess what it means but not for everyone it will be clear.

Response 10: Explanation  has been included the first time it cited in the text: “Figure 5. Traffic intensity 2019-2020 (Q1 and Q2: first and second quarter of the year). (Line 362)

Point 11. Line 329: “new normal.” – dot after quote.

Response 11: Dot after the quote included. (Line 376)

Point 12. Line 348-358: poor format of the text (text alignment).

Response 12: Text aligned according to required format (Line 394-403).

Point 13. Page 12: footnote. Check in the template if the footnote can be used.

Response 13: Footnote has been eliminated (Line 455) and incorporate in the paragraph “Table 3 shows the minimum number of LCVs required to deliver the parcels requested by residents of Madrid Central area, taking into account the market share of the courier sector and delivery productivity based on traffic (interval of 80-125 parcels / LCV is considered following the traffic intensity explained in Appendix I)”. (Line 449-450)

More detail are included in the appendix 1. (Line 625-626)

Point 14. Table 4: this small drawing on left: is it necessary?

Response 14: No, it is not necessary. It is the covid 19 icon to try to help in the visualization of the data, the icon represents the covid period. As it has not had the desired effect, it has been removed

Reviewer 2 Report

Dear author, it was my pleasure to read your paper, as the issues of e-commerce are a highly discussed topic. However, its importance is multiplied in the time of the global pandemic era.

Relationship to literature - the paper demonstrates an adequate use of relevant literature and connects the research questions to the previous literature findings. The authors revise literature on “Urban movement of goods, E-commerce, and urban logistics, Measures and solutions to improve urban logistics”. I would recommend using more current sources in relation to section “E-commerce and urban logistics”, especially actual 2021 research covering the Covid-19 pandemic and its effect on E-commerce in the European area.

Discussion - There is no discussion in relation to existing research results. There is no critical view of the results in comparison with current knowledge in the field. The subject part is not sufficiently developed with regard to the presented research results.

Conclusion - The section on limitations deserves deeper - more specific “treatment”.

Author Response

Thank you for your comments. I really appreciate them and I think they make the paper improve remarkably.

Point 1: Relationship to literature - the paper demonstrates an adequate use of relevant literature and connects the research questions to the previous literature findings. The authors revise literature on “Urban movement of goods, E-commerce, and urban logistics, Measures and solutions to improve urban logistics”. I would recommend using more current sources in relation to section “E-commerce and urban logistics”, especially actual 2021 research covering the Covid-19 pandemic and its effect on E-commerce in the European area.

Response 1: incorporated three new ideas in this section:

  • (Line120-123): (i) the necessary adaptation undertaken by some businesses during lockdown
  • (Line 123-128): (ii) different e-commerce behaviours by various product categories during the pandemic
  • Line (163-174): (iii) evolution of purchasing habits and their implications for urban logistics.

Three current references have been incorporated: (i) E-commerce Europe, 2021 (ii) OECD, 2020 and (iii) Deloitte, 2020.

Point 2: Discussion - There is no discussion in relation to existing research results. There is no critical view of the results in comparison with current knowledge in the field. The subject part is not sufficiently developed with regard to the presented research results.

Response 2: various aspects have been added for each research question, seeking to provide a critical vision, along with a comparison with current knowledge of the subject matter.

These incorporations are:

Line (499-508). Related to the first objective of the paper. “In this context, all methods of transportation were affected and it would be interesting to know how citizens changed their daily commute preferences due to the healthcare crisis. In the case of car access volume, the reduction was similar for total vehicles (67.22%), but significantly lower for LCVs (46.74%). This observation is explained by the fact that, due to the state of alarm, access to the central district was restricted to key activities, including supply of essential goods and services, home delivery of food, healthcare services and the necessary industry to conduct key activities. Under normal conditions, urban distribution accounts for 25% of total traffic, but this percentage rose during the analysed period, due to the decrease in general traffic when compared to LCV traffic reduction. Focusing only on the period of the state of alarm, overall vehicle access to Madrid Central area fell by 82.18% and in the specific case of LCVs, by 67.38%”.

Line (515-521). Related to the second objective of the paper:” It is important to note that this growth of e-commerce was not only due to an increase in shopping frequency by customers who already used the online channel, but also the emergence of new buyers who had previously been reluctant to make purchases through the Internet. The necessity created by the limitations imposed forced these new customers to face that unknown barrier. All the signs seem to indicate that, once this obstacle has been overcome, most new customers will continue making purchases through the new channel”

Line (525-528; 540-541). Related to the third objective of the paper: “Given the large number of couriers with a small market share, one alternative to con-sider would be the consolidation of these operators’ e-commerce parcels through a microhub located in the central district. This could be implemented via public (microhub) and private (logistics operators) collaboration….. A greater decrease in emissions should be sought through the use of other, cleaner means for last mile deliveries (bicycles, delivery on foot, electric vehicles, etc.) or else through greater productivity in the LCV kilometre/delivered parcel ratio. The option of incorporating a microhub to consolidate parcels would also lead to a reduction in CO2 emissions”.

Point 3: Conclusion - The section on limitations deserves deeper - more specific “treatment”.

Response 3: the limitations identified are detailed and specified to greater extent. The paragraph would now be as follows (Line 583-594): “Inevitably, the study has several limitations, which provide valuable paths for future research. First, it would be relevant to have a complete picture of citizen mobil-ity during the pandemic, that is, to know the exact percentage who used public transport, how many used their private vehicles and how many chose to move around on foot or bicycle. This would provide an understanding of the transfers that took place between different methods of transportation. Obtaining this information for the post-Covid period could be very useful in defining better urban mobility and logistics policies in the future. Another essential element would be calculating the exact per-centage of vehicles employed in urban logistics versus total traffic. Understanding this information and knowing delivery schedules would contribute to more efficient and sustainable proposals for urban logistics and traffic in big cities. Furthermore, it would be valuable to evaluate how other modes of transportation delivery (smart lockers, collection points, etc.) may con-tribute to improve economic aspects for couriers and social and environmental aspects for the city.

Round 2

Reviewer 1 Report

Dear Authors,

thank you for taking into account my comments. I hope it helped your to improve your paper. I recommend you paper to be published. I found only 2 small things that should be checked. But these are only problems with formatting. From point of view of the content it is very good.

Be careful, in text after Figure 2, second paragraph, three last lines, you have different type of letters.

Figure 3, probably you covered text with "methodological framework" and it moved a bit.

Check if Figure 5 did not move.

Author Response

Response to Reviewer 1 Comments

Thank you very much for your help. It is an example of how a reviewer can improve the paper. We are very grateful.

Point 1: Be careful, in text after Figure 2, second paragraph, three last lines, you have different type of letters.

Response 1: Thank you. We have changed this paragraph in order to incorporate more current scientific sources. We have incorporated with the correct format (Line 187-190).

Point 2: Figure 3, probably you covered text with "methodological framework" and it moved a bit.

Response 2: we have corrected the text in the figure and checked the place of the figure.

Point 3: Check if Figure 5 did not move.

Response 3: we have checked the place the figure 5 and of the rest of figures in the paper. Thanks.

In addition, once the suggestions and changes have been incorporated, extensive English revision has been carried out by a native-speaking professional translator.

Again, thank you very much for your help.

Reviewer 2 Report

Dear colleagues, 

I would recommend using more current sources in relation to section “E-commerce and urban logistics”, especially actual 2021 research covering the Covid-19 pandemic and its effect on E-commerce in the European area.

By current sources, I mean published research papers, for example in MDPI journals or other CC journals.

Discussion: Please add sources (or explain that it's based on your research) to statements such as and others: "Under normal conditions, urban distribution accounts for 25% of total traffic,..."

Still, there is no discussion in relation to existing research results (comparison of results) from other authors in this field of research. 

Author Response

Response to Reviewer 2 Comments

Dear colleague:

Thank you very much for your help. We really appreciate your comments to improve the paper. We are very grateful.

Point 1: I would recommend using more current sources in relation to section “E-commerce and urban logistics”, especially actual 2021 research covering the Covid-19 pandemic and its effect on E-commerce in the European area.

By current sources, I mean published research papers, for example in MDPI journals or other CC journals.

Response 1: Many thanks. Following your suggestions, we have added the following paragraphs and new sources.

With reference to: 2.1.2. E-commerce and urban logistics:

Line (115-119): “In the United States, the share of e-commerce in total retail sales rose from 11.8 to 16.1% between the first and second quarters; in the United Kingdom from 20.3 to 31.3%. In the EU-27, retail sales via mail order houses or the Internet in April 2020 in-creased by 30% compared to April 2019, while total retail sales diminished by 17.9% (OECD, 2020a).”

Line (126-127): “e.g., by posting products on social media sites and by ordering product pick-up or delivery services (Koch et al., 2020)

Line (136-138): “During the first months of the pandemic, transportation and distribution of goods be-came one of the main causes of disruptions in the supply chain and affected the supply of essential items (Ivanov, 2020; Linton and Vakil, 2020).”

Line (143-144): “In addition, the on-demand economy and its instant deliveries have driven new consumer habits (Dablanc et al., 2017) where q-commerce”…..

Line (187-190): “In this new context, lockers, collection points and mobile warehouses can have a positive impact from various perspectives and for all stakeholders involved in urban logistics by reducing the number of trips, failed deliveries and vehicles required. (Viu-Roig and Alvarez-Palau, 2020).”

Point 2: Discussion: Please add sources (or explain that it's based on your research) to statements such as and others: "Under normal conditions, urban distribution accounts for 25% of total traffic,..."

Response 2: Thank you very much for your help. We have incorporated and updated the source based in this statement and others.

Line 98: (Dablanc, 2011)

Line 503: (DGT, 2020)

Line 512: (see Figure 8). 

Line 515: (ONTSI, 2020). 

Line 571-572:  (Eurostat, 2021; BCG 2020). 

Line 584-587: (Cheval et al., 2020) and (Taniguchi et al., 2020). 

Point 3: Still, there is no discussion in relation to existing research results (comparison of resultsfrom other authors in this field of research. 

Response 3: Thanks again, you are right. Following your considerations, we have discussed and compared the results with existing research as shown below:

Line (492-496): In turn, since the Government implemented the State of Alarm which locked down most of the population until the so-called “new normal” (11/05/2020), road traffic dropped, on average, down to 76.28%. “Similar results were reported in the UK, where road traffic volumes fell by as much as 73% (Budd and Ison, 2020), and in other cities around the world: New York (-74%), Barcelona (-73%), Milan (-74%), Stockholm (-48%) and Sao Paulo (-55%) (year-on-year traffic reduction between 16 March to 22 March, 2020; Statista 2020b).”

Line (571-572): “The exception was e-commerce, where transactions for physical goods increased by 98% during this period, in line with online retail shopping behaviour in other EU countries and the US (Eurostat, 2021; BCG 2020).

Line (580-587): CO2 emissions related to e-commerce last-mile increased 43.1% during the pandemic period, but this increase in CO2 is no relevant if we consider the global reduction of all pollutant emissions in cities due the reduction in traffic and other activities. Average NO2 levels during the week of 16-22 March went down by 41% in Madrid, 51% in Lisbon, 55% in Barcelona, 21% in Milan and 26-35% in Rome (Cheval et al., 2020). Therefore, environmental measures could focus on using innovative technologies: IoT (Internet of Things), big data, parcel lockers, electric vehicles, route optimisation algorithms, collaboration among couriers and the use of urban distribution centres (Taniguchi et al., 2020).

Line (596-598): Focusing on the environmental perspective, the increase in courier activities, added to the new consumption and mobility trends, highlight the need to promote improvements in the current models for urban transport and distribution of goods, including: public-private collaboration for retailers and for transport and logistics operators, environmentally friendly vehicles for city dwellers and raising e-commerce customer awareness and regulation (Russo and Comi, 2020).

In addition, once the suggestions and changes have been incorporated, extensive English revision has been carried out by a native-speaking professional translator.

Again, thank you very much for your help.

Round 3

Reviewer 2 Report

It can be published in the present form.